# Investigation of Trehalose Supplementation Impacting *Campylobacter jejuni* and *Clostridium perfringens* from Broiler Farming

**DOI:** 10.3390/vetsci10070466

**Published:** 2023-07-15

**Authors:** Yang-Chi Fan, Yi-Tei Wu, Yi-Hsieng Samuel Wu, Chia-Lan Wang, Chung-Hsi Chou, Yi-Chen Chen, Hsiang-Jung Tsai

**Affiliations:** 1Zoonoses Research Center and School of Veterinary Medicine, National Taiwan University, No. 1, Sec. 4, Roosevelt Rd., Taipei City 106, Taiwan; d99629005@ntu.edu.tw (Y.-C.F.); vetnicole@gmail.com (C.-L.W.); cchou@ntu.edu.tw (C.-H.C.); 2Department of Animal Science and Technology, National Taiwan University, No. 1, Sec. 4, Roosevelt Rd., Taipei City 106, Taiwan; v5239559@gmail.com (Y.-T.W.); ycpchen@ntu.edu.tw (Y.-C.C.); 3Institute of Food Safety and Health Risk Assessment, National Yang-Ming Chiao Tung University-Yangming Campus, 155, Sec. 2, Linong Street, Taipei 112, Taiwan; wusyhbatu@nycu.edu.tw

**Keywords:** trehalose, tolerance test, *Campylobacter jejuni*, *Clostridium perfringens*, *Lactobacillus*

## Abstract

**Simple Summary:**

Trehalose, a disaccharide, which can be found in animals, plants and micro-organisms, was permitted to be used as a legal food additive by the FDA (USA) and the European Union. Hence, this study aimed to investigate whether trehalose can impact growth performance and pathogenic bacteria inoculation in broilers. In the first experiment, the tolerance of broilers to the addition of trehalose to their feed was investigated. During the 35-day feeding period, it was observed that a trehalose dosage up to 10% does not exert a negative effect on broiler farming. The antibacterial tests of trehalose on *Campylobacter jejuni* and *Clostridium perfringens* were observed over a 5-week feeding period. There was no significant difference (*p* > 0.05) in the bacterial counts of *C. jejuni* and *C. perfringens* in broilers. However, *Lactobacillus* counts significantly increased in these groups with 3% and 5% trehalose supplementation. In summary, trehalose cannot directly impact broilers’ growth performance when *C. jejuni* and *C. perfringens* are found in the broilers’ gut, but it can be used as a prebiotic in their feed by raising *Lactobacillus* counts. Although trehalose did not show promise in reducing *C. jejuni* and *C. perfringens* in poultry, the results of this research are helpful in the study of the pathogen-specific characteristics of trehalose.

**Abstract:**

In 2006, the European Commission banned the use of antibiotic promoters in animal feed. However, there is a new situation in poultry disease where it is necessary to study feed additives, which can overcome the diseases that were previously controlled through the addition of antibiotics and antimicrobial growth promoters in the feed. Therefore, trehalose was investigated to determine whether it impacts the growth performance and pathogenic bacteria (*C. jejuni* and *C. perfringens*) inoculation in broilers. In the first experiment, the tolerance of broilers to the addition of trehalose to their feed was investigated. There was no significant difference (*p* > 0.05) in body weight changes, daily weight gain, feed intake or feed conversion ratio during the feeding period. Within a 35-day feeding period, it was concluded that a trehalose dosage up to 10% does not exert a negative effect on broiler farming. Moreover, there was no significant difference (*p* > 0.05) in the broilers’ growth performance, as well as *C. jejuni* and *C. perfringens* counts in the intestines and feces of broilers observed over a 5-week feeding period. However, *Lactobacillus* counts significantly increased in these groups with 3% and 5% trehalose supplementation. The findings indicate that trehalose supplementation in the feed cannot directly decrease *C. jejuni* and *C. perfringens* counts but may enhance gut health by raising *Lactobacillus* counts in chicken gut, particularly when enteropathogenic bacteria are present.

## 1. Introduction

Animals are known to be major carriers of foodborne pathogens. To mitigate the risk of microbial infectious diseases and improve animal performance, antibiotics and antimicrobial growth promoters have been widely used in animal husbandry for over 50 years, particularly in intensive farming practices. Foodborne pathogens and antibiotics have been reported to impact the human health negatively. Outbreaks linked to foodborne infections have become increasingly complex and challenging due to the emergence of multidrug-resistant (MDR) bacteria, such as *Clostridium* infections and Campylobacteriosis in humans [1,2,3,4,5,6,7,8]. Concerns have been raised about the development of alternative approaches to disease control and prevention in animal husbandry. 

Campylobacteriosis is a leading foodborne bacterial disease across the globe and has held the position as the most frequently reported zoonosis in the European Union since 2005 [9]. The primary causative agents of this disease are *C. jejuni*, with contaminated chicken meat identified as a major source of infection [3]. It is estimated that over 50% of poultry meat worldwide is contaminated with *Campylobacter* [10]. Despite its prevalence, no effective measures have been established to control *Campylobacter* infections in primary broiler chicken production [10]. Once a chicken becomes colonized, the pathogen rapidly spreads, infecting nearly 100% of the flock within just one week [11]. Within a range of feed additives that serve as alternatives to antibiotics, the application of probiotics, prebiotics or synbiotics has been demonstrated to effectively reduce *C. jejuni* bacterial populations in poultry [12]. These effects include inhibiting growth, adhesion and invasion; reducing motility; exhibiting direct antimicrobial activity; and promoting immune function and overall gut health [10,12,13,14,15,16,17,18].

In 2006, the European Commission made the decision to ban the use of commonly used antibiotic promoters in animal feed and to minimize the therapeutic use of antibiotics in animal production. However, the prevalence of *C. perfringens* infection, which causes poultry necrotic enteritis (NE), has created a challenging situation [7,19,20,21]. In the past, the disease was prevented and/or controlled through the addition of in-feed antibiotics and antimicrobial growth promoters [5,8,22]. The ban on the use of these agents as feed additives has resulted in the re-emergence of this disease, which has caused significant economic losses to the global poultry industry [5,20]. This has led to an increase in research focused on new feed additives, including prebiotics, probiotics and other natural compounds, to support animal health and performance while reducing the risk of disease outbreaks and the need for antibiotics [23,24,25,26].

Prebiotics have emerged as a promising alternative to antibiotics in animal production, as they can improve gut health and enhance immune function in a natural and sustainable manner [27,28]. Prebiotics, such as trehalose, were approved as a legal food additive by the FDA in the United States and the European Union. According to current reports [29,30] on the use of trehalose in broiler feed, it is possible that supplementing trehalose instead of antibiotics could result in positive changes to the gut microbiota, improvements in the broiler house environment, a reduction in pathogenic bacteria and a lower incidence of foodborne diseases. In a study conducted by Chen et al., it was found that trehalose has a significant positive effect on the growth of bacteriocin-producing lactic acid bacteria as compared to fructooligosaccharides (FOS) [31]. Furthermore, it has been observed that the addition of trehalose to culture media results in higher bacteriocin production by *Lactobacillus animalis*, *Enterococcus durans* L28-1, *Lactococcus lactis* spp. C101910 and *Lactococcus* sp. GM005 as compared to culture media supplemented with dextrose, FOS and raffinose [31]. These findings suggest that trehalose has potential as a prebiotic, as it can stimulate the growth of beneficial bacteria and enhance their ability to produce bacteriocins, which can help promote gut health and reduce the risk of disease in broiler chickens. It has been observed that the specific activity of trehalose in broiler chickens is not detectable after they reach 21 days of age. This suggests that trehalose may not be efficiently metabolized by older broiler chickens [32]. As a result, alternative strategies may need to be explored to promote the gut health and enhance the immune function of broiler chickens beyond this age [32]. However, it is essential to note that the research on trehalose supplementation in broiler chickens is still limited, and more studies are needed to confirm its potential benefits and determine the optimal dosage. This study was conducted in two parts to investigate the following: (1) the potential tolerance of broiler chickens to trehalose supplementation in their diet; and (2) the effect of trehalose supplementation on the reduction in two common enteropathogenic bacteria (*C. jejuni* and *C. perfringens*) in the digestive tracts of broiler chickens.

## 2. Materials and Methods

### 2.1. Animals and Treatments

In this study, Arbor Acres plus (AA+) broilers, which are widely used in global broiler meat production, were utilized. One-day-old chicks were obtained from a local hatchery (Ju-Ling Farming Co., Ltd., ILan County, Taiwan). The animal protocol used in this study was reviewed and approved by the National Taiwan University Institution Animal Care and Use Committee (IACUU No. 103-008).

For the trehalose tolerance test in broilers, 50 birds were used. In the antibacterial tests of trehalose against enteropathogenic bacteria (*C. jejuni* and *C. perfringens*) in broilers, 75 birds were used to test against each pathogen.

Animals were housed at temperatures of 22–30 °C, relative humidity of 60–70% and a dark–light cycle, which varied according to their age (1 h light/23 h dark for 0–1-week-old birds, 4 h light/20 h dark for birds older than 1 week) as per commercial guidelines. Diets were formulated based on commercial guidelines (Appendix A: Experiment 1; Appendix A: Experiment 2), and water was provided ad libitum during both acclimation and experimental periods.

Broilers were raised in a floor rearing environment with rice chaff bedding, housed in isolation units (approximately 1 m^3^) equipped with high-efficiency particulate-air-filtered air supplies (Dong-Yung Co. Ltd., New Taipei City, Taiwan). The birds were assigned randomly to various experimental groups, and the trehalose product, containing at least 98% pure trehalose dihydrate (C_12_H_22_O_11_ · 2H_2_O), was kindly provided by HAYASHIBARA Co., Ltd. (Okayama, Japan). In Experiment 2, proper procedures were followed to maintain microbial isolation in all groups and prevent cross-contamination.

### 2.2. Preparation of C. jejuni, C. perfringens and L. johnsonii

In this study, two enteropathogenic bacteria, *C. jejuni* (ATCC 700819) and *C. perfringens* (ATCC 13124), were used. The bacterial inocula were prepared based on previous literature and briefly described [33] as follows: *C. jejuni* was cultured in Bolton broth (OXOID Ltd., Hampshire, UK) supplemented with 5% (*v*/*v*) horse blood and incubated for 22 h at 42 °C. Before the oral challenge, the inoculum was subcultured 2–3 times to ensure high bioactivity, and on the day of inoculation, the inoculum was diluted to a concentration of 2.0 × 10^7^ colony-forming unit (cfu)/mL using sterile phosphate-buffered saline (PBS). Final viable bacteria quantification was performed by serially diluting a sample of inoculum and plating it on Columbia blood agar (BD Co., Franklin Lakes, NJ, USA). *C. perfringens* was cultured in brain heart infusion broth (BD Co., Franklin Lakes, NJ, USA) supplemented with 5% (*v*/*v*) horse blood and incubated for 24 h at 37 °C. Before the oral challenge, the inoculum was subcultured 2–3 times to ensure high bioactivity, and on the day of inoculation, the inoculum was diluted to a concentration of 1.0 × 10^9^ cfu/mL using sterile PBS. Final viable bacteria quantification was conducted by serially diluting a sample of inoculum and plating it on blood agar (BD Co., Franklin Lakes, NJ, USA). *L. johnsonii* (ATCC 17474) was also used as reference strain in real-time PCR and was kindly provided by Dr. Chen, Ming-Ju from the Department of Animal Science and Technology at National Taiwan University. It was cultured in *Lactobacilli* MRS Broth (Neogen Co., Lansing, MI, USA) and incubated for 17 h at 37 °C.

### 2.3. Experiment 1: The Tolerance Test of Trehalose on Broilers 

During the 5-week experimental period, fifty AA+ broilers were allocated randomly into five groups (n = 10, Appendix A): (1) control diet, (2) control diet supplemented with 3% (*w*/*w*) trehalose, (3) control diet supplemented with 5% (*w*/*w*) trehalose, (4) control diet supplemented with 7% (*w*/*w*) trehalose and (5) control diet supplemented with 10% (*w*/*w*) trehalose. During this period, the body weight and feed intake were recorded weekly. The birds were monitored daily for the presence of diarrhea and mortality. To prevent potential cross-contamination between groups, the broilers in each group (n = 10) were individually reared in three isolated floor pens. Wood racks and iron nets were used to house the broilers, with 3 or 4 broilers assigned to each pen. Throughout the procedures, strict biosecurity measures were implemented.

### 2.4. Experiment 2-1: The Antibacterial Tests of Trehalose on C. jejuni

In this experiment (Appendix A), *C. jejuni* (ATCC 700819) was orally administered to broilers once they reached the end of their fourth week of feeding (28 days old). A total of 60 broilers (n = 15 per group) were challenged independently with 1 mL of *C. jejuni* at a concentration of 2.0 × 10^7^ cfu/mL [34], with the exception of the control birds (15 birds), which were given 1 mL of sterile PBS. All birds infected with the enteropathogenic bacterium were divided into four groups: (1) no trehalose supplementation, (2) 1% trehalose supplementation, (3) 3% trehalose supplementation and (4) 5% trehalose supplementation. The dosage of trehalose supplementation was determined based on the results of Experiment 1. To prevent potential cross-contamination between groups, the broilers in each group (n = 15) were individually reared in three isolated floor pens. Wood racks and iron nets were used to house the broilers, with 5 broilers assigned to each pen. Throughout the procedures, strict biosecurity measures were implemented. 

### 2.5. Experiment 2-2: The Antibacterial Tests of Trehalose on C. perfringens

In this experiment (Appendix A), *C. perfringens* (ATCC 13124) was orally administered to broilers once they reached 32 days of age. A total of 60 broilers (n = 15 per group), with the exception of the control birds, which were given 1 mL of sterile PBS, were independently challenged orally with 0.5 mL of *C. perfringens* at a concentration of 1.0 × 10^9^ cfu/mL [33]. All birds infected with *C. perfringens* were divided into four groups: (1) no trehalose supplementation, (2) 1% trehalose supplementation, (3) 3% trehalose supplementation and (4) 5% trehalose supplementation. The dosage of trehalose supplementation was determined based on the results of Experiment 1. To prevent potential cross-contamination between groups, the broilers in each group (n = 15) were individually reared in three isolated floor pens. Wood racks and iron nets were used to house the broilers, with 5 broilers assigned to each pen. Throughout the procedures, strict biosecurity measures were implemented.

### 2.6. Sample Collection

Throughout the experimental period, feed intake (n = 3 per group) and body weight changes (n = 10 per group in Experiment 1; n = 15 per group in Experiment 2-1 and 2-2, respectively) were recorded per pen and individually, respectively, weekly. Additionally, the growth performance was further analyzed by previous parameters. In each group, the daily body weight gain, daily feed intake and feed conversion ratio (n = 3) were computed for each broiler pen. One day before inoculation (28th day) and at the end of the experiment (Experiment 2-1 on the 35th day, Experiment 2-2 on the 39th day), sterile cotton swabs were used to collect fecal samples from the cloacal cavity of each broiler for microbial analysis. For microbial analysis, chymus samples (n = 15 per group) were collected from the middle section of the duodenum, jejunum, ileum and cecum after sacrifice.

### 2.7. Isolation and Identification of Bacteria from Chyme and Feces by Real-Time PCR

Four portions of chyme were collected from the duodenum, jejunum, ileum and cecum immediately after sacrifice and stored at −80 °C. Bacterial DNA was extracted from the intestinal chymus samples collected, including the duodenum, jejunum, ileum and cecum (n = 15). Additionally, fecal samples (n = 3) were also collected. The extraction of bacterial DNA was performed using the Favorprep DNA kit (Favorgen Biotech Co., Ping-Tung, Taiwan) by following the manual’s instructions. Real-time PCR was used to detect the presence of different bacteria. The PCR reaction was performed using the SensiFAST HRM Kit (Bioline Reagents Ltd., London, UK) in a two-step cycle. Reactions were carried out in 20 μL PCR mixtures containing 10 μL of 1X SensiFAST^TM^ HRM Mix (Hot-start DNA polymerase, EvaGreen^®^ dye, dNTPs and optimized buffer components, including 3 mM MgCl_2_), 400 nM of each primer and 4 μL of template DNA. Each reaction was performed in three experimental replicates. The reaction involved a 40-cycle reaction with 3 min of polymerase activation at 95 °C, 5 s of denaturation at 95 °C and 30 s of annealing/extension at three different temperature settings (60 °C for *C. jejuni,* 60 °C for *C. perfringens* and 63 °C for *Lactobacillus* spp.). The thermal cycling, fluorescent data collection and data analysis processes were performed using the StepOne™ System (Applied Biosystems) in accordance with the guidelines provided by the manufacturer. The 16S rRNA primers [35] used to detect *C. jejuni* were F: 5′-TCGTGTCGTCAGATGTTGGG-3′ and R: 5′-CGCGGTATTGCGTCTCATTG-3′. The 16S rRNA primers [36] used to detect *C. perfringens* were F: 5′- AAAGATGGCATCATCATTCAA -3′ and R: 5′- TACCGTCATTATCTTCCCCAAA -3′. The 16S-23S rRNA primers [37] used to detect *Lactobacillus* spp. were F: 5′- TGGATGCCTTGGCACTAGGA -3′ and R: 5′- AAATCTCCGGATCAAAGCTTACTTA -3′. Standard curves for the different bacteria were plotted using the pure single strains of *C. jejuni* (ATCC 700819) (range: 10^4^~10^8^ cfu/g), *C. perfringens* (ATCC 13124) (range: 10^3^~10^9^ cfu/g) and *L. johnsonii* (ATCC 17474) (range: 10^9^~10^12^ cfu/g), at different concentrations.

### 2.8. Statistical Analysis

In a completely randomized design (CRD), the treatment groups are formed by randomly assigning the experimental units (broilers). The crucial feature of this design is that every experimental unit has an equal probability of being allocated to any of the treatment groups. This experiment followed a CRD with a significance level set at a 0.05 probability. When a significant difference among groups was identified using one-way analysis of variance (ANOVA), the treatment differences were evaluated using the least significant difference (LSD) test. Statistical significance was defined as a *p* value less than 0.05. All data analyses were conducted using SAS software (SAS Institute Inc., Cary, NC, USA, 2002) with general linear model (GLM) procedures.

## 3. Results

### 3.1. Experiment 1: The Tolerance Test of Trehalose on Broilers

During a tolerance trial of trehalose for broiler chickens, no significant differences (*p* > 0.05) were observed in body weight (BW), daily weight gain, feed intake or feed conversion ratio (FCR) among the groups with 3%, 5%, 7% and 10% trehalose supplementation compared to the control group throughout the test period (Appendix A). In the groups supplemented with 3%, 5% and 7% trehalose, along with the control group, diarrhea was observed, but it only took place during the initial week (0–7 days) of the experiment (Table 1). Based on the veterinarian’s assessment and diagnosis, the diarrhea was attributed to stress rather than being of pathogenic origin. However, no instances of diarrhea were reported throughout the entire trial period for the group with 10% trehalose added.

### 3.2. Experiment 2-1: The Antibacterial Tests of Trehalose on C. jejuni

Throughout the experimental period, body weight changes were not impacted by *C. jejuni* inoculation or trehalose supplementation (Appendix A). Likewise, the average daily weight gain, feed intake and feed conversion ratio showed no significant differences (*p* > 0.05) between *C. jejuni* and trehalose supplementation in both pre-inoculation and post-inoculation periods (Appendix A). This indicates that neither *C. jejuni* inoculation nor trehalose supplementation had an impact on the growth performance of broiler chickens.

To evaluate the reductive effects of trehalose on *C. jejuni*, total counts of *C. jejuni* and *Lactobacillus* were measured using quantitative real-time PCR. Regarding the bacterial counts in various intestinal sections and feces, *C. jejuni* counts in the duodenum, jejunum and ileum were below detectable levels (<10^4^ cfu/g) across all groups, including the control group in the cecum and feces (Table 2). However, *C. jejuni* counts in the cecum and feces reached 10^7.30^~10^7.48^ cfu/g and 10^4.33^~10^4.92^ cfu/g, respectively, with no significant (*p* > 0.05) differences observed among the inoculated groups in these areas.

Considering the total *Lactobacillus* counts in each intestinal section and feces (Figure 1 and Appendix A), there were no significant (*p* > 0.05) differences between the 0% trehalose + C.J. group and the control group (without *C. jejuni* inoculation) in all sections and feces. However, a significant (*p* < 0.05) increase in total *Lactobacillus* counts was observed in the duodenum, ileum and cecum of the 3% trehalose supplementation group. Upon further analysis, the total *Lactobacillus* counts in the duodenum, ileum and cecum of *C. jejuni* inoculated broilers with 3% trehalose supplementation were 251, 8.5 and 37.6 times higher, respectively, compared with those without trehalose supplementation. Conversely, a significant increase (*p* < 0.05) in total *Lactobacillus* counts was found exclusively in the ileum of the group supplemented with 5% trehalose. Further analysis revealed that, in comparison with the group without trehalose supplementation, the total *Lactobacillus* counts in the ileum of broilers inoculated with *C. jejuni* and supplemented with 5% trehalose were 12.58 times greater.

### 3.3. Experiment 2-2: The Antibacterial Tests of Trehalose on C. perfringens

During the experimental period, no significant differences (*p* > 0.05) were observed in body weight changes (Appendix A), average daily weight gain, feed intake and feed conversion ratio among groups during both pre-inoculation and post-inoculation periods (Appendix A). Based on these growth performance indicators, neither *C. perfringens* inoculation nor trehalose supplementation affected broiler growth performance.

Concerning the *C. perfringens* counts in various intestinal sections and feces, the results were similar to those of *C. jejuni*. Bacterial counts in the duodenum, jejunum and ileum were undetectable (<10^3^ cfu/g) among all groups, including the control group in the cecum and feces (Table 3). However, no significant differences (*p* > 0.05) were detected in *C. perfringens* counts in the cecum (10^4.08^~10^4.43^ cfu/g) and feces (10^5.24^~10^5.82^ cfu/g).

Regarding the total *Lactobacillus* counts in different intestinal sections and feces (Figure 2 and Appendix A), a decrease (*p* < 0.05) in total *Lactobacillus* counts was observed in the duodenum and ileum of broilers inoculated with *C. perfringens* compared to the control group (without *C. perfringens* inoculation). However, the counts in other intestinal sections and feces did not significantly differ (*p* > 0.05). Trehalose supplementation at 3% and 5% increased (*p* < 0.05) the total *Lactobacillus* counts in the duodenum and ileum, respectively. In *C. perfringens*-inoculated broilers with 3% trehalose supplementation, the total *Lactobacillus* counts in the duodenum and ileum were approximately 12.6 and 39.8 times higher, respectively, compared to those without trehalose supplementation. Similarly, in broilers inoculated with *C. perfringens* and supplemented with 5% trehalose, the total *Lactobacillus* counts in the duodenum and ileum were approximately 25 and 63 times higher, respectively, compared to those without trehalose supplementation.

## 4. Discussion

### 4.1. Experiment 1: The Tolerance Test of Trehalose on Broilers

Pancreatic amylase breaks down dietary starch into oligosaccharides and/or alpha-dextrins, while disaccharidases at the brush border membrane handle the terminal digestion of these products and the hydrolysis of disaccharides, such as sucrose, trehalose and lactose. Chotinsky et al. [32] found that trehalose activity in chickens decreases dramatically after hatching and is undetectable after 21 days. Consequently, issues such as diarrhea may arise when excessive amounts of some oligofructoses (inulin) [38] or disaccharides (lactose) [39] are consumed due to the lack of digestive enzymes in the small intestine. There is concern that the absence of trehalose in broiler poultry may lead to trehalose negatively impacting the performance of poultry. In this experiment (Appendix A), there were no significant differences (*p* > 0.05) in body weight changes, daily weight gain, feed intake or feed conversion ratio during the feeding period. Yuwares et al. [30] showed that pellet-form basal diets with 0.75% trehalose could also not improve growth performance, but mash-form basal diets with 0.5% trehalose could. They indicated that trehalose is not destroyed during the pellet heating process, but the related bacterial effects may fail. In this study, trehalose was added to feed after the pellet heating process, and the chickens’ growth performance also did not change. As discussed above, the heating process may not be the major cause of the decreased growth-promoting effect, and mash-form feed with trehalose supplementation is suitable for increasing chickens’ growth performance. Additionally, diarrhea (Table 1) and mortality (Appendix A) were only observed in the first week of the experiment across all groups, including the control group without trehalose supplementation. This could be attributed to stress from adapting to a new environment. Based on the results from Experiment 1, it can be concluded that trehalose is a reliable feed supplement, exerting no negative effects on broilers’ growth performance, even when constituting as much as 10% of the feed.

### 4.2. Experiment 2: The Antibacterial Tests of Trehalose on C. jejuni and C. perfringens

It is widely recognized that *C. jejuni* inhabits the avian gut as a commensal organism, with colonized broilers harboring substantial amounts of bacteria in their ceca (typically ranging from 10^6^ to 10^8^ cfu/g), which serves as the primary site for colonization [40]. Within a range of feed additives that serve as alternatives to antibiotics, the application of probiotics, prebiotics or synbiotics has been demonstrated to effectively reduce *C. jejuni* bacterial populations in poultry [12]. The mechanism behind the reduction in *C. jejuni* remains unclear; however, multiple effects of probiotics and prebiotics on *C. jejuni* have been observed. These effects include inhibiting growth, adhesion and invasion; reducing motility; exhibiting direct antimicrobial activity; and promoting immune function and overall gut health [10,12,13,14,15,16,17,18]. In this study, 2 × 10^7^ cfu *C. jejuni* was inoculated in chickens. Then, *C. jejuni* was predominantly present in the ceca, and its concentrations ranged from 10^7.30^ to 10^7.48^ cfu/g, which are similar to *C. jejuni* naturally inhabiting the avian gut. None of the trehalose-supplemented groups showed a significant reduction in *C. jejuni* bacterial counts in the ceca or feces. In this experiment, the results revealed that adding up to 5% trehalose to poultry feed does not provide significant assistance in controlling *C. jejuni* counts within the gastrointestinal tract. Moreover, broilers were infected with *C. jejuni* with bacterial counts in the cecal contents typically ranging from 10^6^ to 10^8^ cfu/g. An amount of 2.0 × 10^7^ cfu *C. jejuni* was orally administered to broilers in order to mimic the real phenomenon. The limit of detection (LOD) in this experiment was 10^4^ cfu/g, and it was theoretically sufficient. The results showed that some segments of the intestines yielded “not detected” (ND) results, indicating that broilers infected with *C. jejuni* were atypical. The delicate effect of trehalose in the amount of *C. jejuni* was probably missed with the higher LOD, and more experiments are needed to study this effect. 

The addition of 3% and 5% trehalose can enhance *Lactobacillus* counts in the chicken gastrointestinal tract, with 3% trehalose supplementation being recommended considering the benefits of feed supplementation. Prebiotics exert their influence by modifying the composition of gut bacteria and the metabolites they produce. They serve as an energy source, fostering the proliferation of probiotics, such as *Bifidobacterium longum*, *L. fermentum* and *L. brevis* [41]. The functionality of probiotics is driven by their ability to produce and stimulate the host’s immune system and their generation of antimicrobial substances. Interestingly, certain prebiotics have the capacity to curb the propagation of harmful pathogens in the gut by strategically intervening in their pathogenic processes [41]. Chen et al.’s research revealed that trehalose exhibits substantial prebiotic capabilities [31]. According to the result, as a broiler supplement, trehalose improves gut health by increasing *Lactobacillus* probiotics counts, not inhibiting the growth of *C. jejuni* and *C. perfringens*. Numerous probiotic studies have found that *Lactobacillus* probiotics can effectively reduce the number of *C. jejuni* in the gut of poultry [13,15,17,42]. In these investigations, selected probiotic strains were incorporated into the feed to enhance the growth performance of broiler chickens while reducing the amount of *C. jejuni* in their gastrointestinal tracts. Other studies involving *Lactobacillus* have demonstrated their inability to significantly decrease *C. jejuni* counts in the gut; however, they did contribute to boosting the immune response [42,43]. It was found that a probiotic blend of *L. acidophilus*, *Bacillus subtilis* and *Enterococcus faecium* did not significantly decrease the cecal colonization of *C. jejuni* in broiler chickens [43]. A recent study revealed that of the 117 strains within the *Bacillus* and *Lactobacillus* genera, only 26 exhibited in vitro inhibitory activity against *C. jejuni* [44]. As a result, the native *Lactobacillus* strain in chicken intestine raised by trehalose could not be helpful in reducing the number of *C. jejuni*, but specific *Lactobacillus* strains could. Therefore, trehalose and selected *Lactobacillus* strains should be a good strategy for chicken resistance to *C. jejuni* infection. Although no direct reduction in *C. jejuni* counts was observed in the groups with added trehalose in this experiment, trehalose acts as a good prebiotic in feed and can increase the amount of *Lactobacillus* in the chickens’ guts. In addition, trehalose can indirectly promote gut health in poultry and potentially reduce *C. jejuni* levels by combining with specific *Lactobacillus* strains. Further experiments should be conducted to confirm this.

It is generally believed that intestinal *C. perfringens* levels of 10^5^ to 10^8^ colony-forming units (CFU/g) or even higher can cause significant damage to the gut. In contrast, *C. perfringens* levels below 10^5^ CFU/g in the gut contents are less likely to cause damage, with clinical symptoms being mild or normal [45,46]. Previous studies have shown that *C. perfringens* concentrations of 10^7^ to 10^8^ CFU/g are required to cause severe clinical symptoms in chickens [47]. In this experiment, chickens were inoculated with 5 × 10^8^ cfu *C. perfringens* to mimic the NE disease model. At the end of the experiment, the *C. perfringens* counts in the cecum decreased from 10^4.08^ to 10^4.43^ cfu/g in all groups. Therefore, there are no NE symptoms and no significant differences in growth performance in chickens. This discrepancy may be due to the fact that the experimental environment lacked co-existing microbial communities and other stress factors, such as invasion from other pathogenic micro-organisms [33,48,49]. Therefore, the inoculated *C. perfringens* did not develop an NE disease model, and the group inoculated with *C. perfringens* did not exhibit a significant difference in growth efficiency compared to the control group.

There are numerous studies demonstrating that the utilization of probiotic micro-organisms, prebiotic substrates that enhance specific bacterial populations or synbiotic combinations of prebiotics and probiotics can effectively mitigate the positive impact of *C. perfringens* in poultry [22,24,50,51]. Another study showed that the use of synbiotics can improve gut health without significantly reducing the counts of *C. perfringens* [51]. In this experiment, the counts of *C. perfringens* in the cecum and feces were not significantly different (*p* > 0.05) among the inoculated groups with or without trehalose supplementation. Moreover, *Lactobacillus* counts decreased in the duodenum and ileum of broilers inoculated with *C. perfringens*, but significantly increased counts were observed in the groups with 3% and 5% trehalose supplementation. The findings from this experiment indicate that 3% trehalose supplementation in the feed cannot directly decrease *C. perfringens* counts but can enhance gut health through raising *Lactobacillus* counts in chicken gut when enteropathogenic bacteria are present. 

## 5. Conclusions

According to this study, trehalose was a safe feed supplement for chickens, and there were no adverse effects, even at a trehalose concentration of up to 10%. However, trehalose did not directly promote chicken growth performance in all groups. In the field, *C. jejuni* and *C. perfringens* co-exist with complex microflora in poultry gut, and in this study, the environment is too simple. As a result, the disease model in chicken is not observed, and the growth performance of all groups is good. According to a previous study [30], mash-form basal diets with 0.5% trehalose improved growth performance in broilers. The non-significant effect of trehalose on growth performance in this experiment is probably limited by diet form.

According to current research, trehalose or trehalose derivatives play important roles in the pathogenicity of various Gram-positive and Gram-negative pathogens. Additionally, trehalose and its derivatives also have significant implications for host colonization and growth, regulating the interactions with the host’s defense mechanisms. In this experiment, 3% trehalose supplementation in the feed can enhance gut health by raising *Lactobacillus* counts in chicken gut when *C. jejuni* and *C. perfringens* exist. A recent study revealed that specific *Bacillus* and *Lactobacillus* strains have in vitro inhibitory activity against *C. jejuni* [44]. Although no direct reduction in *C. jejuni* counts was observed in the groups with added trehalose in this experiment, trehalose acts as a good prebiotic in feed and can increase the amount of *Lactobacillus* in chickens’ guts. Trehalose combination with specific *Lactobacillus* strains may be the strategy to reduce specific pathogens, and more experiments should be conducted in the future.

Trehalose-related effects are typically pathogen-specific [52], and the mechanism of trehalose in broilers is not clear. Therefore, the various studies on trehalose are important for figuring out the mechanism and use strategies. In this study, trehalose did not directly reduce *C. jejuni* or *C. perfringens*, but it enhanced gut health by raising *Lactobacillus* counts in chickens’ guts, particularly when *C. jejuni* and *C. perfringens* were present. Although trehalose did not show promise in reducing *C. jejuni* and *C. perfringens* colonization in poultry, trehalose still had the probability of acting as an additive in broiler diet to control other pathogenic bacteria. Additionally, the result of this research is helpful for further study of the pathogen-specific characteristics of trehalose.

## Figures and Tables

**Figure 1 vetsci-10-00466-f001:**
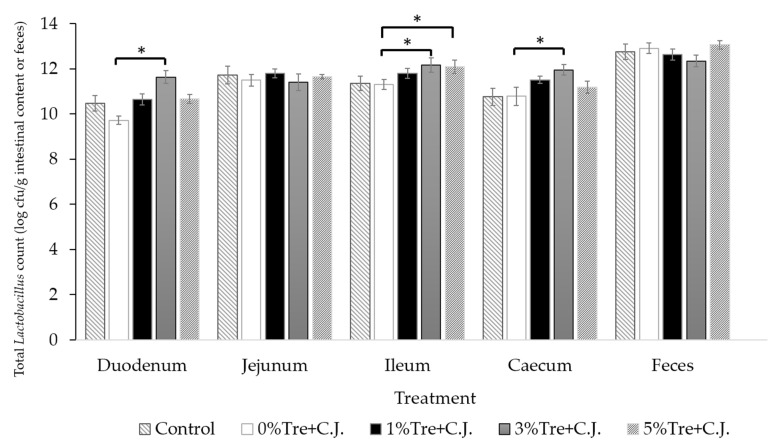
Effects of trehalose (Tre) on total *Lactobacillus* counts in portions of intestine or feces of broilers orally challenged with *C. jejuni* (C.J.). * Asterisks denote that the mean values within each tested portion of intestine or feces are significantly different (*p* < 0.05).

**Figure 2 vetsci-10-00466-f002:**
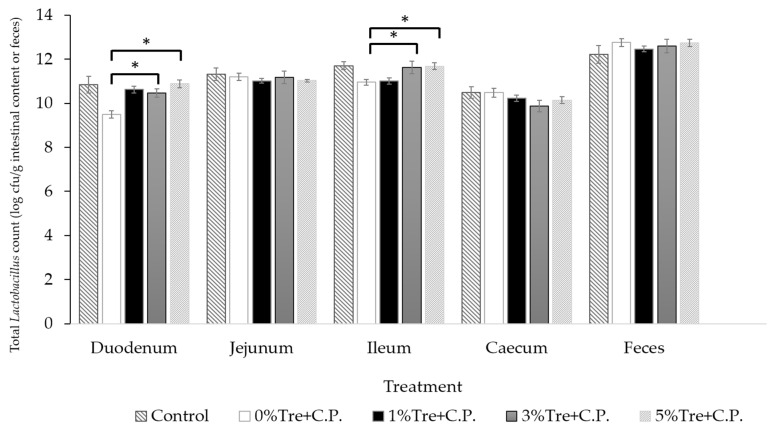
Effects of trehalose (Tre) on total *Lactobacillus* counts in portions of intestine or feces of broilers orally challenged with *C. perfringens* (C.P.). * Asterisks denote that the mean values within each tested portion of intestine or feces are significantly different (*p* < 0.05).

**Table 1 vetsci-10-00466-t001:** Effects of trehalose (Tre) on the incidence of diarrhea phenomenon of broilers in each feeding period and overall feeding period.

	Treatment				
Feeding Period (Day)	Control	3% Tre	5% Tre	7% Tre	10% Tre
0–7	2	3	3	1	0
8–14	0	0	0	0	0
15–21	0	0	0	0	0
22–28	0	0	0	0	0
29–35	0	0	0	0	0
Overall (0–35)	2	3	3	1	0

**Table 2 vetsci-10-00466-t002:** Effects of trehalose (Tre) on *C. jejuni* (C.J.) counts in portions of intestine or feces of broilers orally challenged with *C. jejuni* (C.J.).

Intestinal	Treatment				
Segment/Feces	Control	0%Tre + C.J.	1%Tre + C.J.	3%Tre + C.J.	5%Tre + C.J.
	*C. jejuni* Counts (log cfu/g Intestinal Content or Feces)				
Duodenum	N.D.	N.D.	N.D.	N.D.	N.D.
Jejunum	N.D.	N.D.	N.D.	N.D.	N.D.
Ileum	N.D.	N.D.	N.D.	N.D.	N.D.
Cecum	N.D.	7.33 ± 0.16	7.42 ± 0.13	7.48 ± 0.16	7.30 ± 0.14
Feces	N.D.	4.65 ± 0.17	4.92 ± 0.36	4.55 ± 0.36	4.43 ± 0.25

The data are given as mean ± SEM (n = 14~15, except feces n = 3 with at least triplicate per cage). Mean values within each tested portion of intestine or feces with different letters are significantly different (*p* < 0.05). No significant (*p* > 0.05) differences were observed among the inoculated groups in various intestinal sections and feces. N.D.: not detectable, below 10^4^ cfu/g.

**Table 3 vetsci-10-00466-t003:** Effects of trehalose (Tre) on *C. perfringens* (C.P.) counts in portions of intestine or feces of broilers orally challenged with *C. perfringens* (C.P.).

Intestinal	Treatment				
Segment/Feces	Control	0%Tre + C.P.	1%Tre + C.P.	3%Tre + C.P.	5%Tre + C.P.
	*C. perfringens* Counts (log cfu/g Intestinal Content or Feces)				
Duodenum	N.D.	N.D.	N.D.	N.D.	N.D.
Jejunum	N.D.	N.D.	N.D.	N.D.	N.D.
Ileum	N.D.	N.D.	N.D.	N.D.	N.D.
Cecum	N.D.	4.08 ± 0.16	4.28 ± 0.12	4.43 ± 0.15	4.13 ± 0.23
Feces	N.D.	5.82 ± 0.20	5.50 ± 0.32	5.24 ± 0.33	5.67 ± 0.41

The data are given as mean ± SEM (n = 13~15, except feces n = 3 with at least triplicate per cage). Mean values within each tested portion of intestine or feces with different letters are significantly different (*p* < 0.05). No significant (*p* > 0.05) differences were observed among the inoculated groups in various intestinal sections and feces. N.D.: not detectable, below 10^3^ cfu/g.

## Data Availability

The data and analyses presented in this paper are freely available from the corresponding author upon reasonable request.

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
