# Peer review of "Investigation of Trehalose Supplementation Impacting Campylobacter jejuni and Clostridium perfringens from Broiler Farming"

_vetsci, 2023, doi:10.3390/vetsci10070466_

Round 1

Reviewer 1 Report

The manuscript with the title “Investigation of trehalose supplementation impacting  Campylobacter jejuni and Clostridium perfringens from broiler farming” describes the use of Trehalose additive for poultry nutrition. 

Although the scope of the project is fairly good the methodology of the paper is narrow and there is a lack of data for supporting the usefulness of the additive.  The results do not permit a specific understanding of usefulness of the specific feed and the real implication for preventing the growth of any pathogen.

 The results and conclusions are not supported by the analyses.

The Authors describe significative results regarding the amount of Lactobacillus. Nevertheless, there is a complete lack of info regarding the general composition of the microbiota and it is not clear from the tables if the differences are really significative. The absence of such info does not permit a proper interpretation of the data.

The use of the PCR with missing info on the number of replicates do not permit a complete trusting of the data. LOD seems also too high for Campy detection

Moreover the Authors did not perform microbiological culturing to confirm the results.

Author Response

Dear reviewer,

Thank you very much for your comments. Please refer to the attachment to see my response to your review.

Best regards,

Yang-Chi, Fan

Reviewer 2 Report

Throughout the manuscript several grammatical errors are detected. 

Author Response

(The authors gave the same response as above.)

Reviewer 3 Report

The manuscript is nicely done and written. The study design is appropriate and apparently, the analyses were carefully performed.  I believe that the results are valuable for the scientific community and has significant scientific merit, as it will probably ignite many further studies in the near future.

However, some points need to be clarified before the publication. 

Line 82 – The abbreviation AGPs is used only one time. I see no sense for this acronym.

Line 94 – reference number should follow the authors names – “Tomasik and Tomasik [31-33]”.

Line 129 – Please explain what FOS stands for.

Line 233 – Anatomically “gastrointestinal duct” does not exist. Please use the correct name.

Line 237 – Chyme is a semifluid mixture made of partially digested food and digestive juices formed in the stomach and intestines. It seems that in this study chyme was collected from the intestine only but not from the stomach. The question is why not from the stomach? By the way, plese write just “chyme” not “gastrointestinal chyme”

Line 258 – what was post-hoc test used for ANOVA?

Author Response

(The authors gave the same response as above.)

Reviewer 4 Report

Introduction

The section is long and verbose and also includes a lot of well-known facts. It must be shortened to allow the readers to catch up with the more interesting parts of the manuscript.

Procedures

Can you please described the procedures for selection of the animals and for inclusion into the study? What criteria and what procedures were used?

How did you choose to use these two particular Camp. Strains? Please provide their clinical history and please provide the full genome of the strains please.

The design of the experiments must be summarised in a table.

2.7. ALL the details of the PCR please, not just the primers.

Results

Please move some of the table to supplementary material and please include graphs to enhance the visual presentation for easier understanding by readers.

Discussion

This is rather shallow and not convincing.

Overall: the work is not convincing.

Extensive editing of English language required

Author Response

(The authors gave the same response as above.)

Round 2

Reviewer 4 Report

The authors have answered the comments but did not incorporate the changes into the revised version..............

Moderate editing of English language

Author Response

Dear reviewer, 

Thank you for your comments. Kindly refer to the attached document to review my response in detail.

Best regards,

Yang-Chi, Fan
